# An Early Block in the Replication of the Atypical Bluetongue Virus Serotype 26 in *Culicoides* Cells Is Determined by Its Capsid Proteins

**DOI:** 10.3390/v13050919

**Published:** 2021-05-15

**Authors:** Marc Guimerà Busquets, Gillian D. Pullinger, Karin E. Darpel, Lyndsay Cooke, Stuart Armstrong, Jennifer Simpson, Massimo Palmarini, Rennos Fragkoudis, Peter P. C. Mertens

**Affiliations:** 1The Pirbright Institute, Pirbright, Surrey GU24 0NF, UK; gillian.pullinger@btinternet.com (G.D.P.); karin.darpel@pirbright.ac.uk (K.E.D.); lyndsayscooke@outlook.com (L.C.); jennifer.simpson@pirbright.ac.uk (J.S.); 2Institute of Infection and Global Health, University of Liverpool, Liverpool Science Park IC2, Liverpool L3 5RF, UK; sarmstro@liverpool.ac.uk; 3MRC-University of Glasgow Centre for Virus Research, Sir Michael Stoker Building, Garscube Campus, University of Glasgow, Glasgow G61 1QH, UK; Massimo.Palmarini@glasgow.ac.uk; 4Edinburgh Genome Foundry, University of Edinburgh, Edinburgh EH9 3BF, UK; R.Fragkoudis@ed.ac.uk; 5School of Veterinary Medicine and Science, University of Nottingham, Sutton Bonington Campus, Nottingham LE12 5RD, UK; Peter.Mertens@nottingham.ac.uk

**Keywords:** arbovirus, bluetongue, BTV-26, atypical serotype, reverse genetics, cell binding, VP2

## Abstract

Arboviruses such as bluetongue virus (BTV) replicate in arthropod vectors involved in their transmission between susceptible vertebrate-hosts. The “classical” BTV strains infect and replicate effectively in cells of their insect-vectors (*Culicoides* biting-midges), as well as in those of their mammalian-hosts (ruminants). However, in the last decade, some “atypical” BTV strains, belonging to additional serotypes (e.g., BTV-26), have been found to replicate efficiently only in mammalian cells, while their replication is severely restricted in *Culicoides* cells. Importantly, there is evidence that these atypical BTV are transmitted by direct-contact between their mammalian hosts. Here, the viral determinants and mechanisms restricting viral replication in *Culicoides* were investigated using a classical BTV-1, an “atypical” BTV-26 and a BTV-1/BTV-26 reassortant virus, derived by reverse genetics. Viruses containing the capsid of BTV-26 showed a reduced ability to attach to *Culicoides* cells, blocking early steps of the replication cycle, while attachment and replication in mammalian cells was not restricted. The replication of BTV-26 was also severely reduced in other arthropod cells, derived from mosquitoes or ticks. The data presented identifies mechanisms and potential barriers to infection and transmission by the newly emerged “atypical” BTV strains in *Culicoides*.

## 1. Introduction

Bluetongue virus (BTV) is the causative agent of ‘bluetongue’ (BT), a haemorrhagic disease of domesticated and wild ruminants, although camelids and certain species of large carnivores can also become infected [1,2,3]. Bluetongue is of high economic importance to the agriculture sector, causing fatalities and production losses in livestock, and is notifiable to the World Organization for Animal Health (OIE) [4].

Bluetongue virus is the type species of the genus *Orbivirus*, family *Reoviridae*. It has an icosahedral capsid, approximately 80 nm in diameter, composed of three concentric layers of proteins [5,6]. Twenty-four distinct BTV serotypes (BTV-1 to BTV-24) have been recognised for many decades, the identity of which is determined by the specificity of neutralising antibody interactions with their outer-capsid protein VP2 [7]. These ‘classical’ BTV strains are arthropod-borne viruses (arboviruses), transmitted between their ruminant hosts primarily by adult *Culicoides* biting-midges (Diptera: *Ceratopogonidae*), although more rarely, transmission can also occur via transplacental transmission, or ingestion of infected material [8,9,10].

BTV infects and replicates in its insect vectors, after ingestion as part of a blood meal from an infected vertebrate host, resulting in internal dissemination and infection of the insect salivary glands. This allows the virus to be transmitted to other hosts through the insect’s saliva during subsequent blood meals. Consequently, the incidence of BTV infections is directly linked to the global and seasonal distribution of vector-competent, adult *Culicoides* populations. Currently, 1357 species of *Culicoides* have been described [11], of which at least 13 different species have been implicated as BTV vectors (reviewed in [12]).

The geographic spread of BT outbreaks has recently expanded northwards, beyond their historical distribution (between latitudes 40° S and 53° N), with the virus becoming endemic in southern and central Europe [13,14]. These changes have been linked to increased travel/trade and particularly the effects of climate change on the distribution and seasonal activity of vector species/populations in the region.

Since 2008 several BTV strains have emerged that belong to previously unrecognized serotypes, (including BTV-25, 26 and 27), as well as strains that have not yet been assigned to specific serotypes [15,16]. Some of these novel viruses are adapted to small ruminants (sheep and goats), causing asymptomatic infections, but are unable to replicate effectively in adult *Culicoides sonorensis*, or in KC cells [17,18,19,20]. These atypical strains are believed to be transmitted primarily by direct animal to animal contact [18,20,21].

The ability of an arthropod vector to become infected and subsequently transmit the virus to a susceptible host determines its vector competence. This ability varies between *Culicoides* species, and different levels of susceptibility to BTV infection are also observed between individual insects of the same species, as well as between different strains of BTV [22,23,24,25]. This indicates that host, virus and vector characteristics (genetic factors), as well as different environmental variables can all be involved in determination of transmission efficiency, vector competence of individual insects and the overall vector capacity of vector populations [26,27]. Some aspects of *Culicoides*-dependent vector-competence have been studied (reviewed in [28]) and the role played by several of the viral proteins in the replication of BTV in *Culicoides* cells has been shown in vitro and in vivo [19,29]. However, in general, the determinants that control vector competence and the mechanisms involved are poorly understood.

The genome of BTV comprises ten linear segments of double stranded (ds) RNA, identified as segment 1 to 10 (Seg-1 to Seg-10) in order of decreasing molecular weight [6], which collectively encode seven structural proteins (VP1 to VP7), and at least four non-structural proteins (NS1 to NS4), plus a putative NS5 protein encoded by an additional ORF in segment 10 [30,31,32]. Within the particle, each of the ten packaged dsRNA segments is associated with a transcriptase complex (TC) that consist of an RNA-dependent RNA polymerase (VP1/Seg-1), a helicase (VP6/Seg-9) and a capping enzyme (VP4/Seg-4) [5,33,34]. The innermost capsid shell (‘sub-core’), which surrounds and interacts with both the genome segments and TCs, is composed of 120 copies of VP3 (encoded by Seg-3). The intermediate ‘core surface’ layer contains 260 trimers of VP7 (encoded by Seg-7), which interact with both the outer-capsid and the sub-core. The external ‘outer-capsid’ layer is assembled from 60 trimers of VP2 (encoded by Seg-2) and 120 trimers of VP5 (encoded by Seg-6).

In mammalian cells, the BTV outer-capsid is responsible for cell attachment, via the interaction of VP2 and a still unknown receptor(s) on the cell surface, followed by internalization/cell-membrane penetration mediated by VP5 [35,36,37,38]. The mechanisms by which BTV initiates infection of *Culicoides* cells have been studied to a much lesser extent than in mammalian cells. In addition to intact BTV virus-particles, particles with protease-cleaved VP2 (infectious sub-viral particles—ISVP) and core particles that have lost their outer capsid components, can also be highly infectious for the KC cell line (derived from *C. sonorensis*) and for adult *Culicoides* via an oral route [39,40]. This suggests the use of multiple cell-attachment and entry mechanisms, involving VP2/VP5 or VP7, also potentially utilizing different cell-surface receptors.

VP2 and to a lesser extent VP5, are the most variable of the BTV proteins [7]. The amino-acid/nucleotide sequences of VP2/Seg-2 can be divided into distinct phylogenetic clades, which correlate with BTV serotype [7]. To date, at least 27 distinct serotypes have been recognized, which include the novel, ‘atypical’ BTV strains of BTV-25, -26 and -27 [17,41,42]. Several additional putative novel BTV serotypes have also recently been identified [15,43,44].

The ability of BTV-26 from Kuwait (isolate KUW2010/02) to replicate efficiently in mammalian cells (for example BSR cells), but not in *Culicoides* cells, provides an opportunity to study BTV-vector interactions. Reassortant viruses generated by reverse genetics, containing a selection of genome-segments from BTV-26 and the reference strain of BTV-1, were generated and used in our previous study to identify four genome segments of BTV-26 that completely (Seg-1, encoding VP1—RNA-dependent-RNA-polymerase; Seg-2, encoding VP2—outer-capsid protein; and Seg-3, encoding VP3—sub-core shell protein); or partially (Seg-7, encoding VP7—outer-core protein) restrict replication in vitro in *Culicoides* derived KC cells [19], while maintaining complete ability to infect and replicate in a mammalian BSR cell line. Here we build on our previous work to initially investigate the ability of BTV-26 to replicate in cells derived from other ‘potential’ arthropod-vectors, including both mosquitoes and ticks as well as bovine host derived endothelial cells.

As cell binding and entry is a key factor in restricting virus host-cell range we, furthermore, specifically investigate the role of the outer capsid proteins of BTV-26 in preventing infection of *Culicoides*-derived cells and adult vector *Culicoides*. We report binding and replication studies using BTV-26 and a reassortment virus based on BTV-1, but containing capsid proteins VP2, VP5 and VP7 of BTV-26, and identify that early stages of the BTV-26 replication cycle, including cell-attachment, are impeded in KC cells, but not in mammalian-derived cells. This block cannot be overcome by higher MOIs or temperatures. Like the BTV-26 wild-type, the reassortant virus is also incapable of replicating to transmissible levels in adult female *C. sonorensis* midges, confirming that the capsid of BTV-26 is defective in infecting midges, and greatly contributes to the overall inability of BTV-26 to efficiently replicate and ‘amplify’ in *Culicoides*-derived cells or adult midges.

## 2. Materials and Methods

### 2.1. Viruses and Cells

The viruses used in this study were obtained from the dsRNA virus collection at The Pirbright Institute (TPI) (Table 1). A ‘reverse engineered’ virus (rBTV-1) was generated in BSR cells (a clone of baby hamster kidney cells BHK-21 [45], and obtained from Dr Mark Boyce, The Pirbright Institute, Woking, UK), as already described [19], using mRNAs derived from the reference strain for BTV serotype 1 (BTV-1, isolate (RSArrrr/01), originally from South Africa). The reference strain for BTV-26 (KUW2010/02), (referred to as BTV-26), was isolated from ovine blood samples from Kuwait [41]. The strain, rBTV-1_26 S2,S6,S7_ was ‘reverse engineered’ containing genome-segments 2, 6 and 7 (Seg-2, 6 and 7) from BTV-26 (coding for the larger outer-capsid protein VP2, the smaller outer-capsid protein VP5 and core-surface protein VP7 respectively), with the remaining genome-segments (the genetic ‘backbone’) from BTV-1 RSArrrr/01 [19] (Table 1). Virus stocks were generated in BSR cells.

Mammalian BSR and BFA cells (Bovine Foetal Aorta endothelium cells, catalogue number 87022601, European Collection of Authenticated Cell Cultures, Public Health England, Porton Down, Salisbury, UK) were incubated at +37 °C with 5% CO_2_. BSR cells were grown in Dulbecco’s modified Eagle medium (DMEM) GlutaMAX (Gibco^®^ by Life Technologies Limited, Paisley, UK) with 5% heat-inactivated foetal bovine serum (HI-FBS) (Gibco^®^ by Life Technologies Limited, Paisley, UK) and penicillin (100 U/mL)/streptomycin (100 µg/mL) (1% Pen/Strep) (Gibco^®^ by Life Technologies Limited, Paisley, UK). BFA cells were grown in Nutrient Mixture F-12 Ham’s medium (Sigma-Aldrich^®^, Gillingham, UK) with 10% HI-FBS and 1% Pen/Strep.

KC cells, originally derived from 2-day-old *Culicoides sonorensis* midge embryos [46] (cell line maintained at the Pirbright Institute, Woking, UK, and originally provided by Sally Wechsler, USDA Laramie, Laramie, WY, USA [39]), and U4.4 cells (derived from *Aedes albopictus* mosquito larvae [47], obtained from Dr Sue Jacobs, TPI, Woking, UK) were incubated at +28 °C without CO_2_. KC cells were grown in Schneider’s insect medium (Sigma-Aldrich^®^, Gillingham, UK) with 10% HI-FBS and 1% Pen/Strep. U4.4 cells were maintained with Leibowitz’s L-15 medium GlutaMAX™ (Gibco^®^ by Life Technologies Limited, Paisley, UK) with 10% HI-FBS, 2% Tryptose Phosphate Broth (TPB) (Sigma-Aldrich^®^, Gillingham, UK) and 1% Pen/Strep.

Tick cells lines RAE25 (derived from *Rhipicephalus appendiculatus* tick embryos [48]) and HAE/CTVM9 (derived from *Hyalomma anatolicum* tick embryos [49]) were incubated at +32 °C without CO_2_. RAE25 were maintained and provided ready to infect by Dr Lesley Bell-Sakyi, University of Liverpool, Liverpool, UK.

### 2.2. Time-Course Experiments

Replication kinetics experiments were carried out in mammalian, insect and tick cell lines. Except for the tick cells, these experiments were carried out in T25 cm^2^ flasks or 24-well plates (Corning^®^ Costar^®^, Corning Inc., Corning, NY, USA). Cells were seeded and incubated at their corresponding temperature overnight (Table 2). The following day, cells were infected with BTV diluted in serum-free medium at a multiplicity of infection (MOI) of 0.01, 1, or 5, depending on the experiment. Virus was allowed to adsorb at room temperature for one hour. The inoculum was then removed, and cells washed three times with Dulbecco’s phosphate buffered saline (DPBS) calcium and magnesium free (Sigma-Aldrich^®^, Gillingham, UK). Six mL (for T25 cm^2^) or 1 mL (for 24-well plate) of fresh maintenance medium (Table 2) was added in each flask/well. Supernatant samples (250 μL) and/or coverslips were immediately collected, at 0 days post infection (0 dpi). For infections carried out in T25 cm^2^ flasks, 250 μL of fresh ‘replacement’ medium was added. For infections carried out in 24-well plates, one well was used per time point, and cells were then incubated at appropriate temperatures (Table 2). Supernatant samples were collected at selected times pi and stored at +4 °C until the end of the experiment, when RNA was extracted from 100 µl of each sample using a Kingfisher™ Flex robot (Thermo Fisher Scientific, Swindon, UK) and the LSI MagVet™ Universal Isolation Kit (Thermo Fisher Scientific, Swindon, UK) as per manufacturer’s instructions. Virus copy numbers were quantified using a BTV qRT-PCR assay targeting genome-segment 9 [50], with a 10-fold serial dilution of Seg-9 RNA in vitro transcripts as standard. Coverslips were fixed for 1 h in 4% paraformaldehyde (Santa Cruz Biotechnology, Inc., Dallas, TX, USA) immediately after collection and kept in PBS at +4 °C until processed for immunofluorescence staining.

For characterization of virus replication in tick-derived cells, the cells were seeded in 5.5 cm^2^ flat-sided tubes (Nunc™, Thermo Fisher Scientific, Swindon, UK) (Table 2) and infected at a MOI of 1. Virus inoculum was left to adsorb at room temperature for 90 min and then removed. Cells were washed twice with PBS and 1.5 mL of fresh maintenance medium was added (Table 2). Supernatant samples (250 μL) were collected immediately (0 dpi sample) from each tube and fresh replacement medium was added (250 μL). Tubes were then incubated at +32 °C without CO_2_ for thirty days. Supernatant samples (250 μL) were collected daily during the first 10 days and subsequently every five or ten days, up to thirty days pi. Medium was replaced (250 μL) after each sampling. To maintain cell viability, two-thirds of the medium was changed every 10 days. When a medium change was carried out, a sample was collected before and after the medium change. All samples were stored at +4 °C until the end of the experiment, when viral RNA was extracted, and viral genome copies were quantified (as above).

### 2.3. Virus Titration

End-point titration of virus stocks or selected supernatants were carried out on BSR cells in 96 well tissue culture plates. Briefly, 100 μL of BSR cells in DMEM containing 1% FBS (HI) at a density of 3 × 10^5^ cells/mL was added to each well. In a separate 96 deep-well plate, a 10-fold serial dilution (from 10^−1^ to 10^−8^) of the virus sample to be titrated was made in DMEM/1% FBS (HI)/1% Pen/Strep. 100 μL of the 10^−1^ to the 10^−8^ virus dilution series were then transferred into the 96 well plate with cells (rows A to H). Each sample was tested in quadruplicate (i.e., four columns) and one extra column contained only media (uninfected control). Cells were incubated for 6 days at +37 ± 1 °C with 5% CO_2_ and the number of infected wells with CPE was recorded. Tissue Culture Infective Dose (TCID_50_) values of 50% were calculated using the Karber formula [51]. To calculate MOIs (MOI = PFU/N° of cells), a PFU/mL working estimate was calculated using the formula, 1 TCID_50_/mL = 0.7 PFU/mL.

### 2.4. Virus Purification

Purification of rBTV-1 and BTV-26 was adapted from [52]. Briefly, three T175 cm^2^ flasks (Greiner Bio-One, Monroe, NC, USA) of BSR cells were infected with BTV when cells were 80%-90% confluent and incubated at +37 °C and 5% CO_2_. When 90 to 100% of cytopathic effect (CPE) was observed (usually at around 48–72 hpi), cells were harvested and pelleted by centrifugation at 1912× *g* and +4 °C for 10 min. Supernatants were then discarded, and cell pellet re-suspended in 10 mL of ice-cold 0.5% Triton-TNE buffer with 1× protease inhibitor cocktail (Sigma-Aldrich^®^, Gillingham, UK). A cytoplasmic extract was made using a hand-held glass homogeniser, then centrifuged for 10 min at 1000× *g*, at +4 °C. Five mL of the extract supernatant was kept on ice until required. The remaining 5 mL of extract was mixed with another 5 mL of 0.5% Triton-TNE with 1× Protease Inhibitor Cocktail and the pellet re-suspended. Manual homogenisation and the collection steps were repeated twice more, giving a final volume of collected cytoplasmic extract of 20 mL. The pellet was discarded after this. The cytoplasmic extract was layered on a sucrose gradient consisting of 40% *w/v* sucrose with 0.1% N-Lauroyl Sarcosine (NLS)/0.2M Tris-HCl pH 8.0 on top of a 66% *w/w* sucrose solution, and centrifuged for 2 h in SW28 open tubes (Beckman Coulter, High Wycombe, UK) at 25,000 rpm (rotor Ti32), +4 °C and lowest brake settings. Virus particles were collected from the interface between the two gradients and re-suspended with 0.1% NLS/0.2 M Tris-HCl pH8.0 buffer. Virus purity and viral titres (3.88 × 10^7^ PFU/mL for rBTV-1 and 1.7 × 10^8^ PFU/mL for BTV-26) was confirmed by 10% SDS-PAGE visualised by Pierce silver staining (Thermo Fisher Scientific, Swindon, UK) and endpoint titration, respectively. Virus preparations were stored at +4 °C until needed.

### 2.5. Protein Identification by Mass Spectrometry

Sucrose purified BTV-26 was mixed with 3× blue loading dye containing DTT (New England BioLabs^®^, Ipswich, MA, USA) as per manufacturer’s instructions. The virus-dye mixture was then heated at 95 °C for 5 min before running it in a 4–20% Criterion™ TGX™ Precast Midi Protein gel (Bio-Rad, Hercules, CA, USA) for 2 h as per manufacturer’s instructions. Viral protein bands were visualised by Coomassie^®^ blue (Thermo Fisher Scientific, Swindon, UK), Pierce silver staining or SYPRO^®^ Ruby Protein Gel Stain (Thermo Fisher Scientific, Swindon, UK). The two bands closest to the expected molecular size of VP2_BTV26_ were then excised and kept in ultrapure water at +4 °C until sequenced by mass spectrometry. Briefly, bands were destained and in-gel tryptic digestion performed essentially as described by [53]. Extracted peptides were reduced to dryness using a centrifugal vacuum concentrator (Eppendorf, Hamburg, Germany) and re-suspended in 3 % (*v/v*) methanol, 0.1 % (*v/v*) TFA for analysis by MS. LC-MS/MS analysis was similar to that described by [54]. Peptides were analysed by on-line nanoflow LC using the Ultimate 3000 nano system (Dionex/Thermo Fisher Scientific, Swindon, UK) coupled to a Q-Exactive mass spectrometer (Thermo Fisher Scientific, Swindon, UK). Spectral data were analysed using the PEAKS studio X+ software (Bioinformatics Solutions Inc., Waterloo, ON, Canada, [55]). Tandem MS data were searched against BTV protein database in NCBI and a contaminant database (cRAP, GPMDB, 2012). Search parameters were as follows; precursor mass tolerance set to 15 ppm and fragment mass tolerance set to 0.02 Da. Two missed cleavage events were permitted. Carbamido-methylation (C) was set as a fixed modification and oxidation (M) and acetylation (N-term) were set as variable modifications. The search was semi-specific. The false discovery rate was set at 1%.

### 2.6. Fluorescent Microscopy

Intracellular immune labelling of the viral non-structural protein 2 (NS2) was used to detect and assess the progress of virus replication over time. Fixed cells on coverslips were washed three times with calcium and magnesium free Dulbecco’s phosphate buffered saline (DPBS) and permeabilised for 15 min with 0.1% Triton (Sigma-Aldrich^®^, Gillingham, UK) in DPBS. Cells were subsequently incubated with 0.5% bovine serum albumin (BSA)/DPBS for 30 min at room temperature, followed up by one hour incubation with *Orbivirus* reference antibody 1 (Orab1), in-house, rabbit polyclonal antibodies (raised against bacterial expressed, purified NS2 of BTV-1) diluted 1:2500 in 0.5% BSA/DPBS. Cells were washed three times with DPBS, before incubating them for one hour with goat anti-rabbit IgG Alexa Fluor^®^ 488 secondary antibody (Life Technologies Limited, Paisley, UK), diluted 1:250 in 0.5% BSA/DPBS. After another three washes with DPBS, cellular nuclei were stained with 4′, 6-Diamidino-2-phenylindole (DAPI) (Life Technologies Limited, Paisley, UK) for ten minutes. Finally, coverslips were washed in distilled water, mounted on glass slides on a drop of Vectashield^®^ mounting medium (Vector Laboratories, Burlingame, CA, USA), sealed with nail varnish and analysed by confocal microscope (Leica SP8 CLSM, Leica Microsystems, Wetzlar, Germany).

The percentage of infected cells was calculated by counting at least five-hundred cells (nuclei) per area and assessing for NS2 immunofluorescence. In total, three random areas per coverslip were selected, and two independent experiments were carried out.

### 2.7. Binding Assay

The ability of BTV-26 to bind KC cells was assessed by immuno-labelling of cell surface-bound virus. BTV-1 and BSR cells were used as positive controls. Briefly, BSR (1 × 10^5^ cells) or KC cells (1 × 10^6^ cells) were seeded on 13 mm glass coverslips in 24 well tissue culture plates (Corning^®^ Costar^®^, Corning Inc., Corning, NY, USA) and incubated overnight at +37 or +28 °C, respectively. Cells were then placed on ice and washed once with ice-cold calcium and magnesium free DPBS. Three hundred µL of purified virus (BTV-26 or BTV-1) was added at a MOI of 50, and cells were incubated on ice for one hour to allow virus binding to the cell, but not internalization. Inoculum was then removed and cells washed three times with ice-cold DPBS before fixing them with 4% PFA for one hour. Fixed cells were then immuno-labelled as described for intracellular immuno-labelling, but without membrane permeabilization, using primary antibodies targeting virus particles (structural proteins). In-house guinea-pig Orab279 polyclonal antibody (anti-BTV-1) or goat GT01 polyclonal antibody (anti-BTV-26) were used, diluted 1:2000 or 1:1000 in 0.5% BSA/DPBS, respectively. Cells were washed three times with DPBS, before incubating them for one hour with goat anti-guinea pig IgG Alexa Fluor^®^ 488 or donkey anti-goat IgG Alexa Fluor^®^ 488 secondary antibodies (Life Technologies Limited, Paisley, UK), diluted 1:250 in 0.5% BSA/DPBS. After another three washes with DPBS, cellular nuclei were stained with DAPI for ten minutes. Phalloidin conjugated to Alexa Fluor^®^ 633 dye (Life Technologies Limited, Paisley, UK) was used to stain actin filaments, as per manufacturer’s instructions. Finally, coverslips were washed, mounted and analyzed by confocal microscopy as described above.

### 2.8. Transmission Electron Microscopy (TEM)

KC cells at a density of 7 × 10^5^ cells were seeded on 13 mm Thermanox plastic coverslips (Thermo Fisher Scientific, Swindon, UK) and incubated overnight at +28 °C. Cells were then infected at a MOI of 5 with BTV-1, BTV-26 or rBTV-1_26 S2,S6,S7_ then incubated at +28 °C. At two days pi cells were fixed in phosphate buffered 2% glutaraldehyde (Agar Scientific Ltd., Stansted, UK) for 1 h followed by 1 h in aqueous 1% osmium tetroxide (Agar Scientific Ltd., Stansted, UK). The samples were dehydrated in an ethanol series; 70% for 30 min, 90% for 15 min and 100% three times for 10 min each. A transitional step of 10 min in propylene oxide (Agar Scientific Ltd., Stansted, UK) was undertaken before the samples were infiltrated with a 50:50 mix of propylene oxide and epoxy resin (Agar Scientific Ltd., Stansted, UK) for 1 h. After a final infiltration of 100% epoxy resin for 1 h, the samples were embedded and polymerised overnight at 60 °C. Eighty µm thin sections were cut, collected onto copper grids (Agar Scientific Ltd., Stansted, UK) and grid stained using Leica EM AC20 (Leica Microsystems, Wetzlar, Germany) before being imaged at 100 kV in a FEI Tecnai 12 TEM with a TVIPS F214 digital camera (FEI Company, Hillsboro, OR, USA).

### 2.9. Oral Infection of Adult Culicoides sonorensis Midges

In-house laboratory-reared adult *Culicoides sonorensis* midges [56] were fed on BTV-infected blood meals, as previously described [18,25]. Briefly, approximately 200 two to three day old adult female *C. sonorensis* biting midges were deprived of sugar for 24 h, before being allowed to feed on defibrinated horse blood (TCS Biosciences, Botolph Claydon, UK) containing 10^6^ or 10^7^ TCID_50_/mL of BTV, using the Hemotek system (Hemotek Ltd., Blackburn, UK) with a Parafilm membrane (Cole-Parmer, Vernon Hills, IL, USA). Immediately after infection/feeding (0 days post infection (dpi)), 10 membrane-fed *C. sonorensis* from each group were individually homogenised in 1 mL of serum-free Schneider’s insect medium (Sigma-Aldrich^®^, Gillingham, UK) using 3 mm round metal beads and a Qiagen Tissue Lyser (Qiagen, Hilden, Germany). The remaining blood-fed insects were incubated for 8 days at +25 ± 1 °C with access to 10% sucrose diluted in water (on cotton wool pads). At the end of the incubation period, individual *Culicoides* midges were homogenized and RNA was extracted from 100 µL of midge homogenate using a Kingfisher™ Flex robot and the LSI MagVet™ Universal Isolation Kit (Thermo Fisher Scientific, Swindon, UK) as per manufacturer’s instructions. Virus levels were measured using a BTV qRT-PCR assay targeting Seg-9 [50]. Vector competence rates (proportion of midges with transmissible levels of viral RNA) were calculated as the proportion of midges with significantly higher amounts of viral RNA at 8 dpi, compared to 0 dpi [25,29,57].

### 2.10. Statistical Analysis

For all experiments, results were plotted using the GraphPad Prism software version 7.00 (GraphPad Software, San Diego, CA, USA). The specific statistical test carried out is described in each figure legend. For oral infection of adult *C. sonorensis* midges, sample size was calculated prior to the experiments, using a test for two proportions using Minitab, version 18 (Minitab LLC, State College, PA, USA). Size sample calculated was 95 individuals for each group, with the following assumptions: power value = 0.8; baseline *p* = 0.18; comparison *p* = 0.05 and significance level = 0.05. Since data from vector competence studies was not normally distributed, non-parametric tests were chosen to analyse the data. Comparison of two populations was carried out with Mann Whitney test, while for comparison of multiple groups, Krustal-Wallis test was used. Vector competence rates were estimated with 95% confidence intervals using a one-sample proportion test.

## 3. Results

### 3.1. BTV-26 KUW2010/02 Is Unable to Replicate in Arthropod-Derived Cells In Vitro

The inability of BTV-26 to infect and replicate in adult *Culicoides sonorensis*, or the *C. sonorensis*-derived KC cell line, which are both susceptible to infection by classical BTV strains [18,19,58], led us to ask whether BTV-26 could be vectored by other arthropod species. Our previous study showed low levels of replication of BTV-26 in immune-deficient C6/36 mosquito cells [19]. Therefore, we also assessed the ability of BTV-26 to replicate in several other arthropod-derived cell lines, including U4.4 cells (derived from *Aedes albopictus* mosquitoes), RAE25 cells (from *Rhipicephalus appendiculatus* ticks) and HAE/CTVM9 cells (from *Hyalomma anatolicum* ticks). Host-relevant Bovine Foetal Aorta endothelium cells (BFA) were included for comparison, as a positive control for effective viral replication.

Both rBTV-1 and BTV-26 replicated effectively in bovine BFA cells (Figure 1A) plateauing at approximately 10^9^ genome copies/mL between 72 and 96 hpi. Infection of the mosquito cell line U4.4 with rBTV-1 (Figure 1B) resulted in exponential virus replication from 2 to 7 dpi, which peaked at approximately 10^9^ genome copies/mL. In contrast, there was an overall decrease in BTV-26 genome copy numbers in these mosquito cells (Figure 1B) by the end of the experiment at 10 dpi, compared to the viral inoculum at 0 hpi. Infection of the tick cell lines RAE25 and HAE/CTVM9 with rBTV-1, or BTV-26 (Figure 1C) showed no increase in either infectivity or genome copy numbers during the first 10 days pi. Media changes at 10 and 20 dpi led to some fluctuations of BTV RNA detection in the collected supernatants (Figure 1C), although overall amounts of viral genome copies still decreased between 0 and 30 dpi for both BTV-1 and BTV-26 in the tick cell lines RAE25 and HAE/CTVM9. Overall, these results show that BTV-26 is not able to successfully propagate in these arthropod-derived cell lines.

### 3.2. Cell Binding Studies of Purified BTV-26 in Culicoides-Derived Cells

To elucidate the mechanism/s that restrict BTV-26 infection and propagation in arthropod cells, we next started interrogating the replication cycle of BTV-26 in cells derived from vector-competent midges (KC cells). The ability of purified BTV-26 virus particles to attach to *Culicoides* (KC) and mammalian (BSR) cell-surfaces was first investigated.

Initially, the integrity of purified BTV-1 and BTV-26 particles was tested by electrophoresis in 10% SDS-PAGE gel. The protein migration patterns of purified virus particles (Figure 2A), showed a band at the size of the sub-core protein VP3 of BTV-26 (94 kDa), but not at the predicted size of outer-capsid and cell-attachment protein VP2 (111 kDa) of BTV-26, suggesting that either VP2 and VP3 of BTV-26 (VP2_BTV26_ and VP3_BTV26_) co-migrate under these conditions or that VP2_BTV26_ might have detached from the particle during the purification process. However, analysis in a 4%–20% gradient gel (Figure 2B) did show two bands at ~94 kDa. Subsequent mass spectrophotometry on the two bands confirmed that VP2_BTV26_ was present in our purified BTV-26 preparations (Figure 2C), but most likely migrated faster than VP2_BTV1_ and below VP3, even though the predicted molecular weight for both VP2 proteins should be very similar at around 111 kDa, and bigger than their respective VP3 proteins. This also suggests that VP2 and VP3 of BTV-26 co-migrate in 10% gels.

Having confirmed that our BTV-26 had not lost cell attachment protein VP2 during purification, cell-binding experiments were carried out. Detection of virus bound to the cell surface confirmed the ability of BTV-26 capsid proteins to mediate attachment to mammalian BSR cells (Figure 3A). However, BTV-26 particles were not detected on the surface of KC cells in comparable binding experiments (Figure 3B), indicating that they do not bind efficiently to these insect cells. As KC cells are smaller in size than mammalian BSR cells, we demonstrated that visualization of virus particle binding was not affected by the cell size and that binding of rBTV-1 particles to KC cell membranes was readily detectable (Figure 3B).

### 3.3. The Ability of the BTV-26 Capsid to Block Virus Replication in KC Cells

Having found that BTV-26 is deficient in its ability to bind to the surface of KC cells, we next investigated if this was mediated solely by its capsid proteins or if conditions such as temperature and MOI might mitigate the low binding and facilitate viral replication. The outer-capsid (VP2 and VP5) and outer-core (VP7) proteins of BTV play multiple roles at different stages of the BTV replication cycle. These include cell attachment, internalization of virus particles, and release of transcriptionally active core-particles into the cell cytoplasm, as well as later stages such as assembly and release of progeny virus particles [59].

KC cells were infected at a low MOI (0.01 TCID_50_/cell) at 27 °C with parental strains BTV-26 or rBTV-1, and virus release into the supernatant was measured up to 10 days pi. A reverse engineered virus (rBTV-1_26 S2,S6,S7_) containing three major capsid and core proteins, VP2, VP5 and VP7 derived from BTV-26, in a BTV-1 genetic backbone (Table 1, [19]), was also included in these experiments. Although, as previously shown [19], rBTV-1 replicated and was amplified in KC cells, BTV-26 and rBTV-1_26 S2,S6,S7_ failed to propagate effectively, with viral-genome copy numbers increasing only slightly in the supernatant between days 0 and 2 pi, then remaining unchanged for the rest of the experiment (Figure 4). Next, we investigated whether increasing the MOI 100-fold (MOI of 1) would overcome any obstacle to replication and propagation caused by for example, an inefficient cell-entry process. However, at this MOI, BTV-26 and rBTV-1_26 S2,S6,S7_ still failed to replicate efficiently or spread in KC cell cultures at 27 °C, with only a minor increase in supernatant genome copy numbers between 0 and 2 dpi (from ~10^6^ to ~10^7^ copies/mL), which then remained “flat” until 10 dpi (Figure 4).

BSR and KC cell cultures are usually grown at 37 °C and 27 °C, respectively. Infection of KC cells with rBTV-1 at 37 °C resulted in faster replication than at 27 °C, plateauing after 3 dpi instead of at 7dpi (Figure 4). However, when KC cells were infected at 37 °C with BTV-26, or rBTV-1_26 S2,S6,S7_, there was no significant increase in virus copy numbers up to 10 dpi, with kinetics very similar to infections at 27 °C, indicating that both viruses again failed to replicate and amplify successfully.

### 3.4. The Capsid of BTV-26 Blocks the Early Stages of Virus Replication Cycle in KC Cells

BTV non-structural protein 2 (NS2) is synthesized soon after the virus core enters the cell cytoplasm and initiates viral mRNA synthesis [60]. NS2 can therefore be used as a marker for the early stages of replication, indicating that cell-entry, mRNA synthesis and translation have occurred. NS2 was detected at 2 dpi in 4.6% of KC cells infected with rBTV-1 at a low MOI of 0.01, increasing to 67% of cells at 4 dpi (Figure 5A,B), showing that the virus had successfully entered the cells, initiated mRNA and protein synthesis, and was spreading effectively to other cells. In contrast, NS2 was detected in <0.05% of KC cells infected with BTV-26 at either 2 or 4 dpi, confirming that infection and virus replication are severely restricted (Figure 5A,B).

However, low numbers of KC cells did become infected with rBTV-1_26 S2,S6,S7_ at an MOI of 0.01, with NS2 detected at both 2 and 4 dpi (in 1% and 0.75% of the cells, respectively) (Figure 5A,B), suggesting that a low level of cell entry and initiation of viral RNA transcription and protein translation had been achieved, but infection did not progress or spread to additional cells over this time period. With a 100-fold increase of the MOI (MOI of 1 or 5) (Figure 5C), NS2 was still undetected in KC cells infected with BTV-26. However, a slight increase in the number of cells expressing NS2 (2.92 %) was observed at 2dpi with rBTV-1_26 S2,S6,S7_. Together with the results already described (Figure 3 and Figure 4), this indicates that early stages of the viral replication cycle are strongly restricted by the capsid proteins of BTV-26, although if this block is overcome, at least some of the later stages of virus replication can proceed, albeit seemingly not resulting in significant spread of infection to other cells.

Transmission Electron Microscopy (TEM) of pelleted KC cells infected at an MOI of 5 with rBTV-1, BTV-26 or rBTV-1_26 S2,S6,S7_ (Figure 6), confirmed that a few cells were actively infected with rBTV-1_26 S2,S6,S7_ at 2 dpi. These cells contained viral inclusion bodies (VIBs), (composed of NS2), viral ‘tubules’ (composed of NS1) and newly assembled core particles in the cell cytoplasm, demonstrating that viral mRNA/protein synthesis and core particle assembly had occurred. However, whole virus particles were not seen in these infected cells, suggesting that there might be other blocks late in the replication and assembly pathway of rBTV-1_26 S2,S6,S7_ in KC cells, again possibly mediated by VP2 of BTV-26.

VIBs, NS1 tubules, or core particles were not observed in KC cells infected at a high MOI with BTV-26. When internalization of a few BTV-26 particles could occur (as suggested by the data with rBTV-1_26 S2,S6,S7_), both immunofluorescence staining and TEM failed to detect replication of BTV-26. This suggests additional blocks to virus replication mediated by other proteins of BTV-26 such as VP1, or VP3 of BTV-26 [19].

### 3.5. Capsid Proteins of BTV-26 Restrict BTV Replication in Adult C. sonorensis

The biological importance of the in vitro results presented was assessed by orally infecting adult female *Culicoides sonorensis* midges, with blood meals containing rBTV-1, BTV-26 or rBTV-1_26 S2,S6,S7_. Virus replication was measured at 8 days post feeding (dpf), and vector competence ratios were determined as previously described [25].

Initial infections of *C. sonorensis* adults with 10^6^ TCID_50_ of virus/mL of blood (a viral load similar to the average peak viraemia in an infected ruminant) revealed a low competence rate for rBTV-1 (1.39%) (Appendix A), suggesting that the reverse engineered rBTV-1 was a poor positive control for infection. Direct comparison with other BTV-1 strains and conditions of infection (Appendix A) showed that the vector competence of *C. sonorensis* for rBTV-1 was significantly lower than for the reference strain (RSArrrr/01) (identified here as wild type (wt) BTV-1). Interestingly, passage of rBTV-1 in KC cells (rBTV-1 KC) prior to infecting *C. sonorensis* increased viral replication efficiency of this “synthetic” virus within the vector insects. Although beyond the scope of this study, this suggests a role for viral variants and strain-diversity in infection and replication efficiency.

As rBTV-1 (BSR cell passaged) was, however, needed as matched positive control for BTV 26 and rBTV-1_26 S2,S6,S7_, we investigated whether the vector competence rate of *C. sonorensis* for rBTV-1 could be increased by raising the viral dose in the blood meal [39]. Orally infecting *C. sonorensis* midges with a 10-fold increase in the amount of rBTV-1 showed a significant increase in vector competence rate (Appendix A). Therefore, subsequent infections of adult midges with BTV-26 and rBTV-1_26 S2,S6,S7_ were carried out using blood meals containing 10^7^ TCID_50/_mL of virus.

Analysis of midges orally infected with blood containing 10^7^ TCID_50_/mL of rBTV-1 (Figure 7A) showed that 139 out of 140 individuals (99.29%) contained detectable levels of viral RNA (vRNA) on day 8 pf. Midges that were positive for vRNA, could be divided into two significantly different subpopulations, which we identified as “competent” and “non-competent” based on significantly higher or lower amounts of vRNA compared to levels in midges at 0 dpf (Median Ct_competent_ = 19.02 vs: Ct_day0_ = 24.30, *p* ≤ 0.001: Median Ct_non-competent_ = 33.55 vs: Ct_day0_ = 24.30, *p* ≤ 0.001). Twenty-six midges had higher vRNA levels than midges on day 0 pf, indicating that 18.56% (95% CI: 12.50–26.01%) of the midges were vector-competent for rBTV-1.

None of the surviving 103 midges fed with BTV-26 at 10^7^ TCID_50_/mL (Figure 7B) had viral RNA levels at 8 dpf above those at 0 dpf, indicating a competence rate of 0% (95% CI: 0–2.87%). The median Ct for the 10 RNA positive but non-competent midges (Median Ct_non-competent_ = 37.74) was significantly higher (*p* ≤ 0.001) than that on 0 dpf (Ct_day0_= 29.26).

None of the 144 midges that were orally infected with rBTV-1_26 S2,S6,S7_ had levels of vRNA on 8 dpf that indicated virus replication to transmissible levels (Figure 7C). Thirty-six midges had low vRNA levels at 8 dpf, significantly below values on 0 dpf (Ct_day0_ = 28.23, *p* ≤ 0.001), identifying them as non-competent (Median Ct_day8_ = 37.67). This confirms that the capsid proteins of BTV-26 not only restrict viral infection and replication in *Culicoides* cells in vitro, but also restrict BTV replication in adult BTV-competent *C. sonorensis* midges.

## 4. Discussion

The work presented shows that replication of the ‘atypical’ BTV-26 (strain KUW2010/02) is severely restricted in numerous arthropod cells (results summarized in Table 3), including cells derived from the BTV-vector *spp*. *Culicoides sonorensis* (KC cells), as well as in adult *C. sonorensis* midges. We show for the first time that the inability of BTV-26 to complete its replication cycle in KC cells starts with a block in the early stages of its replication, including a greatly reduced ability of the virus to bind to the surface of KC cells. This block is mediated by capsid proteins of BTV-26, preventing amplification and propagation of the virus in vivo within the insect-vector.

The BTV outer-capsid is composed of the two most variable virus proteins (VP2 and VP5), which are involved in cell attachment (via VP2) and penetration of the cell membrane (via VP5), during the early stages of infection by intact virus particles in both mammalian and insect cells [37]. BTV core particles that have lost their outer coat proteins VP2 and VP5 can also initiate cell entry in KC cells, and less efficiently in some mammalian cells (e.g., BHK but not CHO (Chinese Hamster Ovary cells)), mediated by the core-surface protein VP7 [39,52]. Our previous research showed that mono-reassortant BTV-1 viruses containing either VP5 or VP7 derived from BTV-26 can replicate in both KC and BSR cells, while BTV-26 and rBTV-1_26 S2,S6,S7_ can replicate in mammalian BSR cells but not in KC cells (this study and [19]). We therefore conclude that the failure of rBTV-1_26 S2,S6,S7_ to infect and replicate in *Culicoides* cells is most likely due to VP2 of BTV-26.

Although not strictly quantitative and potentially missing low frequency events, confocal microscopy suggests that BTV-26 is largely unable to attach to the *C. sonorensis* cell-membranes, mediated by VP2_BTV26_. This represents a major barrier to cell-entry and infection, preventing productive replication, demonstrated by a failure to detect NS2 synthesis by BTV-26 in KC cell. The cell-receptors for BTV binding have not yet been identified in either mammalian or insect cells, although there is evidence to suggest that BTV uses different entry pathways and therefore possibly different receptors, depending on serotype, target cell and viral particle type [35,36,38,40,52].

However, BTV-26 does bind to permissive mammalian BSR, or bovine BFA cells, in both cases leading to infection and amplification of the virus genome. This confirms that VP2 of BTV-26 is a fully functional mammalian-cell-attachment protein. Protein sequencing by mass spectrometry confirmed the presence of VP2 in purified BTV-26 particles, as well as suggesting that it migrates faster on SDS-PAGE gels than VP2_BTV-1_, even though both proteins have a similar predicted size, and no obvious differences in hydrophobicity and charge (using https://www.expasy.org/resources/protparam, accessed on 25 February 2021). Whether there are still differences in their folding and/or post-translation modifications, which could potentially also explain differences in their cell attachment properties, needs to be further investigated.

A small number of KC cells did become infected with rBTV-1_26 S2,S6,S7_, resulting in viral protein synthesis and at least the early stages of replication and particle assembly, generating VIBs, tubules and core particles, as detected by TEM. In contrast to rBTV-1, fully intact virus particles were not identified in the cytoplasm of rBTV-1_26 S2,S6,S7_ infected KC cells, although the very small percentage of cells infected by this reassortant virus limits the conclusions that can be drawn by TEM. This low level of infection might reflect binding and entry of alternative particle types (e.g., core particles or membrane enveloped virus particles (MEVP) [61]) that are likely to be present in tissue culture-derived virus stocks). In addition to a block in the initial stages of cell binding and entry, the failure of rBTV-1_26 S2,S6,S7_ to spread to other cells in KC cultures might reflect additional blocks to either the final assembly of the outer-capsid of progeny virus particles, or a failure of any intact virus particles generated to be released from infected cells, processes that would both involve VP2 of BTV-26, as well as its effective interactions with internal components/proteins of the insect cells [62,63,64,65].

In summary, our results support the hypothesis that *C. sonorensis* is not able to act as a vector for BTV-26 KUW2010/02, and that BTV-26 might not be a vector-borne virus. Together with previous evidence of direct-contact transmission [18,21], our results provide further critical evidence that direct-contact rather than insect vectors constitute its main route of transmission Further research is required to determine if this applies to other strains of atypical BTV serotypes [20,66] or if they could be vectored by other arthropods, including other *Culicoides* species. VP2 is the most variable BTV protein and the respective VP2s of the BTV-1 and BTV-26 strain utilized in this study only share 38.5% similarity in their aa sequence (GenBank accession numbers BTV-1 VP2: FJ969720.1/BTV-26 VP2: HM590642.1). Furthermore, only recently have some parts of the overall VP2 structure of BTV-1 been modelled and proposed [37,67]. Nonetheless, further work on VP2_BTV26_ could focus on identifying domains, motifs, residues, and cell surface proteins involved in binding of VP2_BTV26_ to mammalian cells and its inability to bind to KC cells [37,67,68]. This could potentially also shed light on viral genetic determinants required for direct-contact transmission.

Results presented here highlight the importance of studying the atypical BTV strains, despite their low pathogenicity, as they can exchange genome-segments (reassort) with conventional BTV strains, generating novel progeny viruses with unpredictable phenotypes. This could potentially result in the emergence of economically significant BTV strains that do not require vectors for transmission, potentially extending their seasonal and geographic range. The atypical BTV strains, “engineered” reassortant viruses and KC cells provide useful tools to address questions relating to BTV/vector interactions, cell tropism, transmission mechanisms and determinants of virulence. With the recent detection of additional atypical serotypes [16,43], their epidemiological importance and occurrence appears to be greater than initially anticipated.

## Figures and Tables

**Figure 1 viruses-13-00919-f001:**
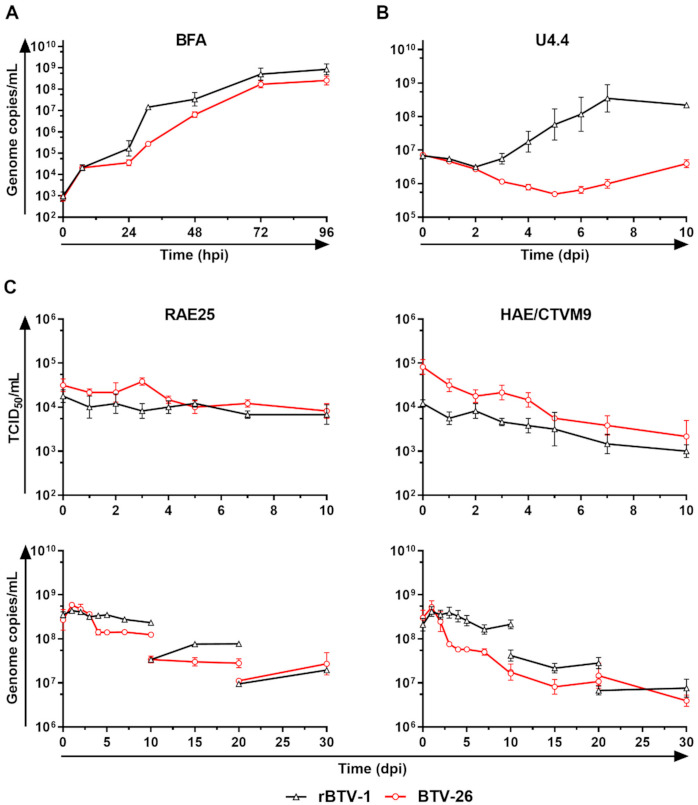
Replication kinetics of BTV-1 and BTV-26 in mammalian and arthropod cell lines. Bovine-BFA cells (**A**) were seeded at a density of 1.00 × 10^5^ cells/well. *Aedes*-U4.4 cells (**B**) were seeded at a density of 2.50 ×10^7^ cells/flask. In (**C**), tick-derived *Rhipicephalus appendiculatus*-RAE25 cells or *Hyalomma anatolicum*-HAE/CTVM9 cells were seeded at a density of 2.40 ×10^6^ cells/tube or 2.00 × 10^6^ cells/tube, respectively. Cells were infected at a MOI of 0.01 (**A**,**B**) or 1 (**C**) with rBTV-1 (black triangles) or BTV-26 (red circles). Infected BFA cells were incubated at +37 °C with CO_2_, while U4.4 were incubated at +28 °C without CO_2_ and tick cells at +32 °C without CO_2_. In all panels, 250 μL of supernatant were collected at each indicated time point. BTV genome copy numbers were determined using the qRT-PCR BTV Seg-9 assay [50] on extracted RNA and the results are expressed as the mean of genome copies/mL of two independent experiments (**A**,**B**) or one experiment in triplicate (**C**) ± standard error of the mean. In (**C**), 50 μL of supernatant were used to calculate 50% tissue culture infective dose (TCID_50_/mL) in BSR cells.

**Figure 2 viruses-13-00919-f002:**
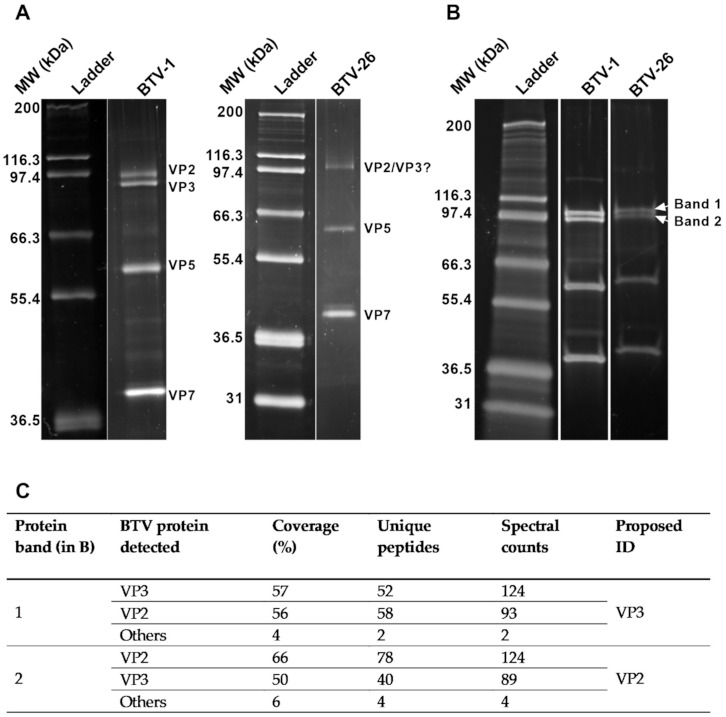
Protein profile of BTV-26 particles and VP2 identification. Purified particles of rBTV-1 and BTV-26 were run in a 10% SDS-PAGE gel (**A**) or a 4%–20% Gradient SDS-PAGE gel (**B**) and stained with SYPRO^®^ Ruby protein gel stain. BTV particles consists of 3 protein layers: the capsid (VP2 and VP5), the outer core (VP7) and the inner core (VP3). In (**A**,**B**), ladder is Mark12™ Unstained Standard. Sequencing of Bands 1 and 2 (in **B**) by mass-spectrophotometry identified them as BTV-26 VP3 and VP2, respectively (**C**).

**Figure 3 viruses-13-00919-f003:**
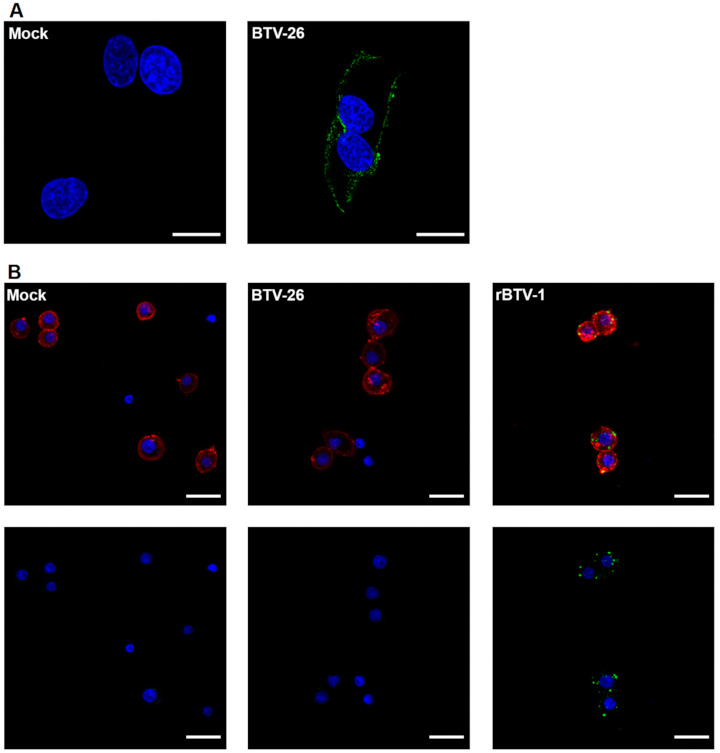
Binding ability of BTV-26 in mammalian and *Culicoides* midge cells. BSR (**A**) or KC (**B**) cells were either mock infected or infected with purified rBTV-1 (**B** only) or BTV-26 (**A**,**B**) at a MOI of 100 (**A**) or 50 (**B**) in ice-cold conditions to prevent virus internalization. After 60 min, cells were washed three times with ice-cold serum-free medium, fixed with 4% PFA and labelled for DNA (DAPI-blue), Actin (Phalloidin-633, red) and BTV-1 (Orab279-Alexa Fluor 488, green) or BTV-26 (GT0128-Alexa Fluor 488, green). Cells were visualized by confocal laser scanning microscopy. In (**A**), the scale bar represents 20 μm for all panels, while in (**B**), the scale bar represents 5 μm, and it is a representative image of 3 experiments.

**Figure 4 viruses-13-00919-f004:**
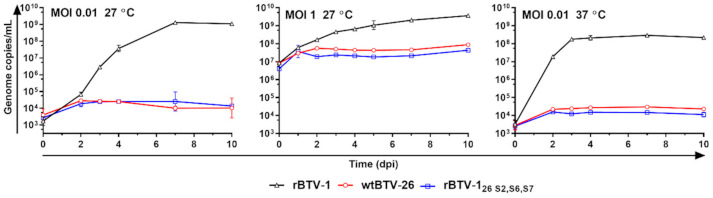
Replication kinetics of rBTV-1, BTV-26 and rBTV-1_26 S2,S6,S7_ in *Culicoides* cells under several conditions. KC cells were seeded at a density of 7.00 × 10^5^ cells/well on coverslips, and infected at a MOI of 0.01 or 1, with rBTV-1 (black triangles), BTV-26 (red circles) or rBTV-1_26 S2,S6,S7_ (blue squares). Infected cells were incubated at +27 °C or 37 °C without CO_2_. Two hundred and fifty μL of supernatant were collected at each indicated time point, and total RNA extracted from 100 μL of supernatant. BTV genome copy numbers were determined using the qRT-PCR BTV Seg-9 assay [50]. Results are expressed as the mean of genome copies/mL of two independent experiments ± standard error of the mean.

**Figure 5 viruses-13-00919-f005:**
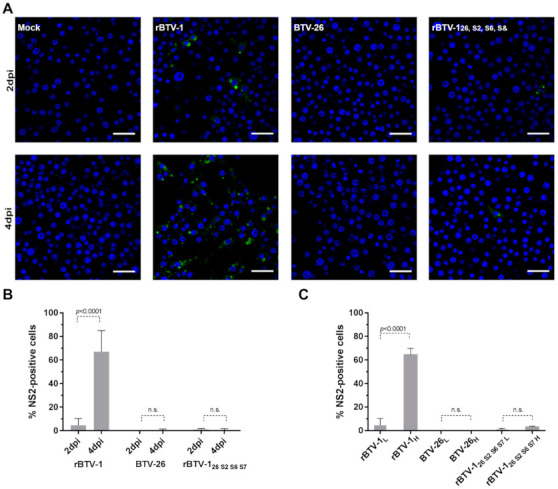
Intracellular replication of rBTV-1, BTV-26 and rBTV-1_26 S2,S6,S7_ in *Culicoides* cells. KC cells were seeded at a density of 7.00E+05 cells/well on coverslips in a 24-well plate at a low MOI of 0.01 (**A**,**B**). At 2 or 4 days pi cells were fixed with 4% PFA and labelled for DNA (DAPI, in blue) and viral NS2 (Orab1-Alexa Fluor 488, in green). Cells were visualized by confocal laser scanning microscopy (**A**) and the percentage of NS2 positive cells from two independent experiments calculated (over 3000 cells counted) (**B**). In (**A**), the scale bar represents 10 μm for all panels. In (**C**), a comparison of the percentage of NS2 positive cells at 2 dpi between KC cells infected at low (L) MOI (0.01) or high (H) MOIs (1–5) is shown. Means were compared by carrying out ANOVA tests: a Sidak’s multiple comparison test (**B**) or a Tukey’s multiple comparison test (**C**).

**Figure 6 viruses-13-00919-f006:**
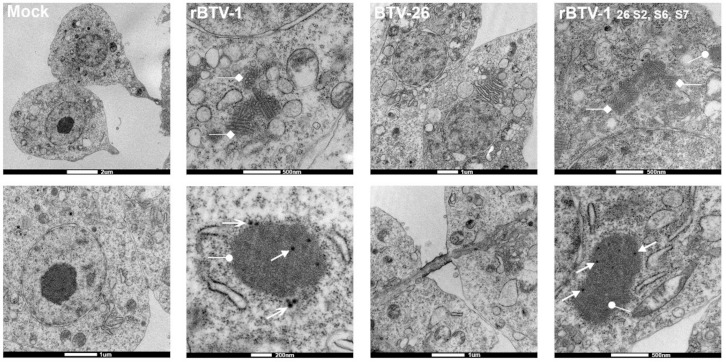
Transmission electron microscopy (TEM) of *Culicoides* cells infected with rBTV-1, BTV-26 or rBTV-1_26 S2,S6,S7_. KC cells on coverslips were either mock infected or infected at a MOI of 5 and incubated at +27 °C with rBTV-1, BTV-26, or rBTV-1_26 S2,S6,S7_. At 2 dpi, cells were fixed with phosphate buffered 2% glutaraldehyde. Cells were visualised by TEM (FEI Tecnai 12 TEM with a TVIPS F214 digital camera). Open arrows show whole virus particles (only in rBTV-1), arrows show viral core particles, ovals show viral inclusion bodies (VIBs) made of NS2, and diamonds show viral microtubules made of NS1.

**Figure 7 viruses-13-00919-f007:**
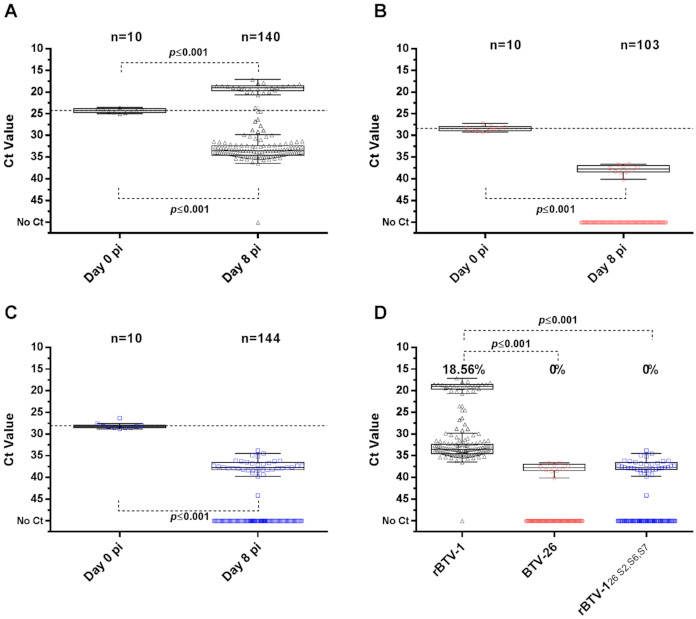
Oral infection of adult *Culicoides sonorensis* midges with rBTV-1, BTV-26 and rBTV-1_26 S2,S6,S7_. Plotted results for 0 and 8 dpi of adult female *C. sonorensis* orally infected with 10^7^ TCID_50_/mL of (**A**) rBTV-1; (**B**) BTV-26; and (**C**) rBTV-1_26 S2,S6,S7_. In (**A**–**C**) dotted lines show the threshold that indicates viral levels of transmissibility/vector competence. Box-and-whiskers show median (line), 1st and 3rd quartiles (25th and 75th percentiles) of the different determined populations (Whiskers were defined by the Tukey method, and comparison of medians by Mann-Whitney test). Comparison of all groups at 8 dpi is shown in (**D**), where calculated competence rates are shown as percentages (2 proportions comparison-Minitab). In all panels each symbol represents one individual midge.

**Table 1 viruses-13-00919-t001:** List of viruses.

Virus (ORC number) ^1^	Cell Passage History ^2^	Origin ^3^	Comments
BTV-1 (RSArrrr/01)	E2, BHK9, BSR1 *	Natural (TPI)	Strain used for segment cloning, identified as wtBTV-1
BTV-26 (KUW2010/02)	E1, BHK2, BSR1	Natural (TPI)	Strain used for segment cloning and infections, identified as BTV-26
rBTV-1 (BTV-RV0023)	BSR3	Synthetic	Rescued strain, derived from BTV-1 RSArrrr/01 cloned segments
rBTV-1_26 S2,S6,S7_ (BTV-RV0020)	BSR3	Synthetic	Rescued strain comprised of wtBTV-1 derived backbone, with genome-segments 2, 6 and 7 derived from BTV-26

^1^ *Orbivirus* Reference Collection (ORC) number. Viruses available in the ORC at The Pirbright Institute (TPI): https://www.reoviridae.org/dsRNA_virus_proteins/ReoID/BTV-isolates.htm (accessed on 14 May 2021). ^2^ Notation of cell passage history: E = passaged in embryonated chicken eggs, BHK in Baby Hamster Kidney cells, BSR in a clone derived from BHK, as described in Section 2.2, together with the number of passages in that cell line/system; ^3^ Synthetic viruses generated in [19] were obtained from the ORC at TPI; * Cell passage history recorded at TPI upon obtaining of this virus isolate.

**Table 2 viruses-13-00919-t002:** Cell culture conditions during virus infection.

Cell Line	No. of Seeded Cells	Incubation (°C)	Maintenance Medium
BSR	1 × 10^5^ cells/well (24 well plate) 3 × 10^6^ cells/flask (T25 cm^2^ flask)	37 ± 1 with 5% CO_2_	DMEM GlutaMAX™ + 1% HI-FBS + 1% Pen/Strep
BFA	1 × 10^5^ cells/well (24 well plate)	37 ± 1 with 5% CO_2_	Nutrient Mixture F-12 Ham’s medium + 1% HI-FBS + 1% Pen/Strep
KC	7 × 10^5^ cells/well (24 well plate) 2.5 × 10^7^ cells/flask (T25 cm^2^ flask)	27 ± 1 without CO_2_	Schneider’s insect medium + 10% HI-FBS + 1% Pen/Strep
U4.4	2.5 × 10^7^ cells/flask (T25 cm^2^ flask)	27 ± 1 without CO_2_	Leibowitz’s L-15 GlutaMAX™ + 10% HI-FBS + 2% TPB + 1% Pen/Strep
RAE25	2.4 × 10^6^ cells/tube (5.5 cm^2^ flat-sided tubes)	32 ± 1 without CO_2_	L-15 GlutaMAX™ + 10% HI-FBS + 5% TPB
HAE/CTVM9	2.4 × 10^6^ cells/tube(5.5 cm^2^ flat-sided tubes)	32 ± 1 without CO_2_	L-15/H-Lac + 20% HI-FBS

**Table 3 viruses-13-00919-t003:** Summary of the ability of rBTV-1 and BTV-26 to replicate in several mammalian and arthropod-derived cell lines.

Cell Line/Organism	rBTV-1	BTV-26	rBTV-1_26S2, S6, S7_
BSR (Hamster–fibroblasts)	+	+	+ ^1^
BFA (Bovine–aorta endothelium)	+	+	NA
KC (*C. sonorensis* midge–larvae)	+	-	-
U4.4 (*A. albopictus* mosquito–)	+	-	NA
RAE25 (*R. appendiculatus* tick–)	-	-	NA
HAE/CTVM9 (*H. anatolicum* tick–)	-	-	NA
Adult *C. sonorensis* midges	+	-	-

“+” indicates efficient replication, whereas “-” indicates severely deficient or no replication at all; ^1^ in [19].

## Data Availability

All data are contained within the article.

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
