# Peer review of "An Early Block in the Replication of the Atypical Bluetongue Virus Serotype 26 in *Culicoides* Cells Is Determined by Its Capsid Proteins"

_viruses, 2021, doi:10.3390/v13050919_

Round 1
Reviewer 1 Report
In this paper, Busquets and co-authors characterize replication blocks of ‘atypical’ bluetongue viruses in insect cells. BTV is an important veterinary and wildlife pathogen that had classically been considered an arbovirus that replicated alternately in insect and vertebrate hosts. Relatively recently discovered ‘atypical’ BTVs, including BTV-26 viruses, seem to be directly transmitted between vertebrate hosts without a vector intermediate. In fact, they don’t replicate in culicoides cells and in previous work, the authors mapped this inability to replicate to 4 of the virus’s 10 genome segments by generating recombinant reassortant viruses that include 6 segments of BTV1, which can replicate in insect cells, and 4 segments of BTV26, which can’t. The 4 segments in question encode the RdRp and 3 structural proteins that make up different parts of the capsid.
In this study the authors follow up on the previous work to attempt to identify the stage of viral replication at which atypical BTVs are blocked in insect cells using a reassortant virus that this time contains 3 segments from BTV-26 that encode 3 capsid proteins: VP2, VP3, and VP7. They also evaluate whether BTV26 can replicate in other arthropod cell lines (Aedes albopictus and some tick line) as a test of the possibility that other arthropods besides culicoides midges might serve as vectors.
In general this is a well written and well done paper. Defining *why* atypical viruses can’t replicate in culicoides cells would represent an interesting advance in the field. The possibility that other arthropods might vector atypical BTVs is a bit of a long shot, but it’s a hypothesis worth testing.
The main way that the paper could be improved relates to the evidence presented in figure 3, which is really the main evidence upon which the authors support the conclusion that virus is blocked at an at attachment. But there are a couple technical issues with this figure.
For one thing, detection of BTV-1 and BTV-26 uses different antibodies, so it is difficult to compare the staining from these different antibodies (and attachment) directly. Fig 3A for some reason does not include a panel corresponding to BTV-1 binding to BSR1 cells. Including such an image (with comparable exposure conditions used throughout) would better enable a side-by-side comparison. The concern is that BTV-26 attachment is occurring in Fig 3B, but the staining and exposure conditions are insufficient to detect the attachment: There is no real positive control since BTV-1 staining uses a different Ab, and the full set of comparisons are not shown in Fig 3A.
Even if staining were believably comparable, it would improve the credibility of this conclusion to include some kind of quantitative measure of the microscopy data. The authors state that “Although not strictly quantitative…” But it would be possible to quantify microscopy images somehow.
Or, the authors could use an alternative cell attachment protocol, for instance one that uses qRT-PCR to quantify relative amounts of bound virus particles (the relative amounts of viral RNA in the particles), as in:
https://doi.org/10.1186/1743-422X-6-217
This would have the advantage of being able to use the same primers for BTV-1 and the reassortant BTV1/26 by targeting one or more of the shared BTV-1 segments.
The authors show that viral replication (RNA levels) and protein synthesis are substantially decreased for BTV-26 and the reassortant virus in figure 4 and 5, but other early but post-attachment/entry blocks would also produce these phenotypes.
Author Response
We thank this reviewer for the review and overall positive assessment of the study. We have considered all suggestions and replied to them separately below:
- Binding of BTV-26 / Figure 3:
Reviewer 1 comments: The main way that the paper could be improved relates to the evidence presented in figure 3, which is really the main evidence upon which the authors support the conclusion that virus is blocked at an at attachment. But there are a couple technical issues with this figure.
For one thing, detection of BTV-1 and BTV-26 uses different antibodies, so it is difficult to compare the staining from these different antibodies (and attachment) directly. Fig 3A for some reason does not include a panel corresponding to BTV-1 binding to BSR1 cells. Including such an image (with comparable exposure conditions used throughout) would better enable a side-by-side comparison. The concern is that BTV-26 attachment is occurring in Fig 3B, but the staining and exposure conditions are insufficient to detect the attachment: There is no real positive control since BTV-1 staining uses a different Ab, and the full set of comparisons are not shown in Fig 3A.
Even if staining were believably comparable, it would improve the credibility of this conclusion to include some kind of quantitative measure of the microscopy data. The authors state that “Although not strictly quantitative…” But it would be possible to quantify microscopy images somehow.
Authors’ response: We highly value the reviewer’s comments and agree that direct and quantitative comparison of BTV-1 and BTV-26 binding is difficult as BTV particle detection requires serotype-specific (and therefore different) antibodies. Therefore, our main aim for this section and Figure 3 was to demonstrate that BTV-26 particles readily bind to mammalian BSR cells (Figure 3A) but not to the Culicoides derived KC cells (Figure 3B) and for this comparison the same antibody against BTV-26 particles was used. However, as KC cells are much smaller in size we wanted to outline that the inability of visualising BTV-26 binding to KC cells was not influenced by the size of the cells, hence we demonstrated that we could successfully visualise binding of rBTV-1 to KC cells.
We therefore propose to change the text in the manuscript slightly to highlight this aim better. We have also changed the order of the panels in Figure 3B from Mock - rBTV-1 – BTV-26 to Mock - BTV‑26 and then rBTV-1.
Original text (starting in line 525 of the revised version): Detection of virus bound to the cell surface confirmed the ability of BTV-26 capsid proteins to mediate attachment to mammalian BSR cells (Figure 3a). Under similar conditions, purified rBTV-1 particles were also identified bound to the surface of KC cells, however BTV-26 was not detected on the surface of KC cells (Figure 3b), indicating that it does not bind efficiently to these insect cells.
New text: Detection of virus bound to the cell surface confirmed the ability of BTV-26 capsid proteins to mediate attachment to mammalian BSR cells (Figure 3a). However, BTV-26 particles were not detected on the surface of KC cells in comparable binding experiments (Figure 3b), indicating that they do not bind efficiently to these insect cells. As KC cells are smaller in size than mammalian BSR cells, we demonstrated that visualization of virus particle binding was not affected by the cell size and that binding of rBTV-1 particles to KC cell membranes was readily detectable (Figure 3b).
- Alternative assay for binding:
Reviewer 1 comments: Or, the authors could use an alternative cell attachment protocol, for instance one that uses qRT-PCR to quantify relative amounts of bound virus particles (the relative amounts of viral RNA in the particles), as in:
https://doi.org/10.1186/1743-422X-6-217
This would have the advantage of being able to use the same primers for BTV-1 and the reassortant BTV1/26 by targeting one or more of the shared BTV-1 segments.
Authors’ response: We are very grateful to the reviewer of highlighting an alternative viral-binding assessment method which we will definitely consider for future work. We do however feel that our immunofluorescence confocal microscopy studies are sufficient for our aim to qualitatively demonstrate that BTV-26 particles are readily detectable binding to BSR but not to KC cells. We therefore feel that the significant work required to design a qRT-PCR assay specifically for the strains used in this study and subsequent validation (for example to proof the absence of free RNA) of the binding assay would not be essential to our aim in this study.
- Results and discussion:
Reviewer 1 comments: The authors show that viral replication (RNA levels) and protein synthesis are substantially decreased for BTV-26 and the reassortant virus in figure 4 and 5, but other early but post-attachment/entry blocks would also produce these phenotypes.
Authors’ response: We fully agree with the reviewer’s comments that other early but post-attachment/entry blocks would also produce the phenotype observed in Figure 5. We have tried to acknowledge the same considerations of a block in the early stages of replication throughout the manuscript. However, taking into consideration our results in figure 3 (cell-attachment) we believe that a reduced ability to bind to insect cells will have an important impact on blocking the early stages of BTV-26 replication, and hence deserves to be specifically highlighted. To better reflect these considerations, we have slightly modified some of our arguments in the manuscript as follows:
In the results section presenting figure 5, we propose the following changes to leave it more general and acknowledge that other early but post-attachment/entry blocks that would also produce these phenotypes:
Original text (Section 3.4, starting in line 611 of the revised version): Together with the results already described (Figures 3 and 4), this indicates that early stages of the viral replication cycle (i.e. viral cell attachment and internalization) are strongly restricted by the capsid proteins of BTV-26, although if entry is achieved, at least some of the later stages of virus replication can proceed, albeit seemingly not resulting in significant spread of infection to other cells.
New text: Together with the results already described (Figures 3 and 4), this indicates that early stages of the viral replication cycle are strongly restricted by the capsid proteins of BTV-26, although if this block is overcome, at least some of the later stages of virus replication can proceed, albeit seemingly not resulting in significant spread of infection to other cells.
In the discussion we suggest the following modification:
Original text (starting in line 715 of the revised version): We show for the first time that the inability of BTV-26 to complete its replication cycle in KC cells starts with a block in the early stages of its replication, mainly due to a reduced ability of the virus to bind to the surface of KC cells.
New text: We show for the first time that the inability of BTV-26 to complete its replication cycle in KC cells starts with a block in the early stages of its replication, including a greatly reduced ability of the virus to bind to the surface of KC cells.
Reviewer 2 Report
This manuscript presents a thoroughly detailed description of a study to verify and expand previous findings of this group that the outer capsid proteins VP2 and VP5 and the inner core protein VP7 of the atypical bluetongue virus serotype 26 are responsible for an early block in infection of cells of the vector Culicoides sonorensis, as well as mosquito and tick cells. These findings provide important information for understanding the mechanisms of BTV vector cell entry and possibly virus assembly and release in vector cells, and are obviously the basis for further research.
Nevertheless, questions arise while reading the manuscript that might be easily answered by the authors with existing data. First, why was the recombinant virus BTV-126 S2,S6,S7 the only recombinant chosen for this study? Very brief and incomplete summaries of a previous study by the authors (PLoS One, 2016) are given in the introduction and discussion. This reviewer suggests that in the final paragraph of the introduction in this manuscript, instead of a thorough summary of the results of the current study, the authors could provide a more complete summary of the previous study to give a rationale for design of the current study. Second, the authors state, correctly, that further studies should focus on identifying domains, motifs and residues of VP2BTV26 involved in cell binding. It would be instructive to provide the parallel amino acid sequence alignment of VP2 proteins from BTV1 and BTV26 in this manuscript to begin this identification.
Minor:
The manuscript should be carefully proofread to correct many minor grammatical errors.
Author Response
We are grateful for the overall positive assessment of the manuscript and thank the reviewer for the additional suggestions for improvement which we have addressed below:
- Introduction:
Reviewer 2 comments: First, why was the recombinant virus BTV-126 S2,S6,S7 the only recombinant chosen for this study? Very brief and incomplete summaries of a previous study by the authors (PLoS One, 2016) are given in the introduction and discussion. This reviewer suggests that in the final paragraph of the introduction in this manuscript, instead of a thorough summary of the results of the current study, the authors could provide a more complete summary of the previous study to give a rationale for design of the current study.
Authors’ response: We thank the reviewer for highlighting that we need to strengthen our description of our current study rationale and its relationship to our previous work. We, therefore, propose the following changes in the introduction to improve the section while highlighting the aims and conclusions of the study:
Original text (starting in line 142 of the revised manuscript): Reassortant viruses generated by reverse genetics, containing a selection of genome-segments from BTV-26 and the reference strain of BTV-1, were used in our previous study to identify four genome segments of BTV-26 that completely or partially restrict replication in vitro in Culicoides derived KC cells [19]. These include: Seg-1, encoding VP1 (RNA-dependent-RNA-polymerase); Seg-2, encoding VP-2 (outer-capsid protein); Seg-3, encoding VP3 (sub-core shell protein); and Seg-7, encoding VP7 (outer-core protein).
We have further investigated the ability of BTV-26 to replicate in cells derived from other ‘potential’ arthropod-vectors, including both mosquitoes and ticks. We report binding and replication studies using BTV-26 and a reassortment virus based on BTV-1, but containing capsid proteins VP2, VP5 and VP7 of BTV-26, which shows a reduced ability to attach to, enter and productively infect insect-vector cells. We identify cell-binding / entry as initial steps in the replication cycle of BTV-26 that are impeded in KC cells, but not in mammalian-derived cells. This block cannot be overcome by higher MOIs or temperatures. Like the BTV-26 wild-type, the reassortant virus was also incapable of replicating to transmissible levels in adult female C. sonorensis midges, confirming that the capsid of BTV-26 is defective in infecting midges, and contributes to the overall inability of BTV-26 to efficiently replicate and ‘amplify’ in Culicoides-derived cells or adult midges.
New text: Reassortant viruses generated by reverse genetics, containing a selection of genome-segments from BTV-26 and the reference strain of BTV-1, were generated and used in our previous study to identify four genome segments of BTV-26 that completely (Seg-1, encoding VP1 - RNA-dependent-RNA-polymerase; Seg-2, encoding VP2 - outer-capsid protein; and Seg-3, encoding VP3 - sub-core shell protein); or partially (Seg-7, encoding VP7 - outer-core protein) restrict replication in vitro in Culicoides derived KC cells [19], while maintaining complete ability to infect and replicate in a mammalian BSR cell line. Here we build on our previous work to initially investigate the ability of BTV-26 to replicate in cells derived from other ‘potential’ arthropod-vectors, including both mosquitoes and ticks as well as bovine host derived endothelial cells.
As cell binding and entry is a key factor in restricting virus host-cell range we, furthermore, specifically investigate the role of the outer capsid proteins of BTV-26 in preventing infection of Culicoides-derived cells and adult vector Culicoides. We report binding and replication studies using BTV-26 and a reassortment virus based on BTV-1, but containing capsid proteins VP2, VP5 and VP7 of BTV-26, and identify that early stages of the BTV-26 replication cycle, including cell-attachment, are impeded in KC cells, but not in mammalian-derived cells. This block cannot be overcome by higher MOIs or temperatures. Like the BTV-26 wild-type, the reassortant virus is also incapable of replicating to transmissible levels in adult female C. sonorensis midges, confirming that the capsid of BTV-26 is defective in infecting midges, and greatly contributes to the overall inability of BTV-26 to efficiently replicate and ‘amplify’ in Culicoides-derived cells or adult midges.
- Discussion - VP2 proteins alignment:
Reviewer 2 comments: Second, the authors state, correctly, that further studies should focus on identifying domains, motifs and residues of VP2BTV26 involved in cell binding. It would be instructive to provide the parallel amino acid sequence alignment of VP2 proteins from BTV1 and BTV26 in this manuscript to begin this identification.
Authors’ response: We thank the reviewer for highlighting that our discussion in this section could be further improved. The outer capsid protein VP2 is the most variable BTV protein and the amino acid identity of VP2 from BTV-1 and BTV-26 is only 38.5%, We therefore believe that addition of the AA sequence alignment between these two proteins will not be very informative within this manuscript, but propose to add the GenBank accession numbers for both proteins and highlight their divergent nature. Furthermore, the three-dimensional structure of VP2 is also still not fully resolved and only a few domains and motifs of VP2 have been identified and characterised to date. The cell binding domain (and the cell receptor) remain unknown for any BTV serotype. We have therefore modified the text in the discussion to qualify our discussion about investigating VP2 of BTV-26 further:
Original text: (starting in line 785 of the revised manuscript): Further work on VP2BTV-26 could focus on identifying the domains, motifs, residues, and cell surface proteins involved in binding of VP2BTV26 to mammalian cells and its inability to bind to KC cells, as well as the VP2 motif / residues of different BTV serotypes and the proteins to which they bind [37, 66, 67]. This could potentially also shed light on viral genetic determinants required for direct-contact transmission.
New text: VP2 is the most variable BTV protein and the respective VP2s of the BTV-1 and BTV-26 strain utilised in this study only share 38.5% similarity in their aa sequence (GenBank accession numbers BTV-1VP2: FJ969720.1 / BTV-26 VP2: HM590642.1). Furthermore, only some parts of the overall VP2 structure of BTV-1 have been modelled and proposed [37, 66]. Nonetheless, future work on VP2BTV26 could focus on identifying domains, motifs, residues, and cell surface proteins involved in binding of VP2BTV26 to mammalian cells and its inability to bind to KC cells [37, 66, 67]. This could potentially also shed light on viral genetic determinants required for direct-contact transmission.
- Minor grammatical errors:
Reviewer 2 comments: The manuscript should be carefully proofread to correct many minor grammatical errors.
Authors’ response: The manuscript has been proofread and several changes in the text have been made as shown by the track changes in the revised manuscript.